# Rethinking Smoking and Quitting in Low-Income Contexts: A Qualitative Analysis with Implications for Practice and Policy

**DOI:** 10.3390/ijerph22071122

**Published:** 2025-07-16

**Authors:** Monique T. Cano, Oscar F. Rojas Perez, Sara Reyes, Blanca S. Pineda, Ricardo F. Muñoz

**Affiliations:** 1Department of Psychiatry, Yale University School of Medicine, New Haven, CT 06511, USA; oscar.rojasperez@yale.edu; 2Department of Psychiatry and Behavioral Sciences, University of California, San Francisco, CA 94143, USA; rmunoz@paloaltou.edu; 3Institute for International Internet Interventions for Health, Palo Alto University, Palo Alto, CA 94304, USA; bpineda@paloaltou.edu; 4University of Nebraska Medical Center, Omaha, NE 68198, USA

**Keywords:** addiction, tobacco use, smoking, poverty, low-income, disparities, social environment, qualitative

## Abstract

Despite a general decline in smoking rates among the U.S. population, smoking among low-income populations remains disproportionately high, likely due to the social determinants of health. To inform tailored approaches and responsive public health policies, the aim of this study was to qualitatively explore the sociocultural contexts, attitudes, and behaviors regarding smoking and quitting in a sample of low-income adults who smoke. In-depth, semi-structured qualitative interviews were conducted with 20 adults. Participants were recruited through local safety-net primary care clinics and community sites. A thematic analytic approach was utilized to analyze transcribed interviews. In exploring smoking and quitting within the context of low-income individuals, the following six themes were identified: caught between health and tobacco use; the nuances of context; roadblocks to quitting; motivation without movement; a temporary escape; and one size does not fit all. Insights into sociocultural and environmental contexts that shape smoking and quitting among low-income individuals revealed a complex interplay of factors that perpetuate smoking behavior and make it difficult to achieve sustained cessation. The study findings point to the importance of patient-centered and collaborative approaches that tailor smoking cessation efforts to the unique needs and lived experiences of low-income people who smoke.

## 1. Introduction

Tobacco-related disparities, including higher rates of preventable smoking-related morbidity and mortality, disproportionately impact low-income people who smoke [1,2]. Despite effective smoking cessation interventions [3] and a general decline in smoking rates among the U.S. population [4], individuals who experience socioeconomic disadvantage continue to smoke more cigarettes at higher rates than people from higher income brackets, have lower self-efficacy for quitting, a more difficult time quitting, and are less likely to successfully quit [5,6,7]. Low-income individuals also often report higher levels of depressive symptoms, and high depression scores are linked to increased smoking rates, reduced cessation success, and a high likelihood of relapse [8]. These disparities are likely the consequence of other interrelated social determinants of health (SDOH) such as lower educational attainment, poorer access to healthcare and quality healthcare services, neighborhood environments, and social contexts [9,10,11,12].

The social and environmental circumstances surrounding low-income people who smoke play a critical role in inhibiting their ability to successfully quit and may also maintain smoking behavior [13,14]. For instance, low-income people who smoke tend to have more friends and family members who smoke [13]. This larger social network of people who smoke can create additional challenges (e.g., triggers and smoking cues when seeing others use tobacco, lack of social support for quitting, social pressure) making smoking cessation more difficult [15]. Social norms within neighborhoods shape smoking behavior, and longitudinal studies have found that individuals living in high poverty neighborhoods are at higher risk for smoking initiation and increased smoking in adulthood [10,16,17]. Similarly, low-income people who smoke are more likely to encounter and access smoking-friendly environments where smoking is allowed, which consequently is linked to smoking more cigarettes [12]. In addition, tobacco industry marketing is highly concentrated in neighborhoods with lower socioeconomic status [18,19], compounding the social and environmental triggers that low-income people who smoke encounter.

Social Cognitive Theory (SCT) provides a framework for understanding what drives smoking behavior, with a particular focus on social learning and environmental interaction [20]. According to SCT, smoking is a multifaceted behavior that is shaped by reciprocal interactions between internal personal factors (cognitive, affective, and biological), behavior, and the environment [20,21,22]. For example, experiencing stress (negative affective state) can drive an individual to seek relief by smoking (behavior) cigarettes, which are disproportionately marketed and more readily available in low-income neighborhoods (environmental triggers and convenience). However, this path could begin anywhere. For instance, living in low-income neighborhoods where tobacco is easily accessible and marketed frequently (environmental convenience and triggers) normalizes tobacco use and, via modeling, influences the initiation and continuation of smoking (behavior), which can then lead to nicotine dependence (biological) and addiction (cognitive/psychological). As such, the components of SCT have been widely used to identify, explain, intervene, and predict factors related to smoking and smoking cessation [20,22]. Given the multifaceted nature of smoking, research has demonstrated that effective interventions must address the various determinants that influence smoking [23]. However, despite advances in successful smoking cessation interventions [3,23], tailored approaches are needed to effectively address tobacco-related disparities among low-income groups [24].

To inform the development of tailored approaches and responsive public health policies, further in-depth research is needed to explore the complex factors that maintain smoking and make cessation difficult among low-income populations [24]. As a result, the research team deemed a qualitative approach to be suitable for the present study, given the methodology’s nature to gain a deeper, more nuanced understanding of behavior and context [25], which is often limited in quantitative research. Grounded in SCT, the primary objective of the current study was to qualitatively explore the sociocultural contexts, attitudes, and behaviors regarding smoking and quitting in a sample of low-income adults who smoke. In the present study, the term “sociocultural” is specific to a low-income context, encompassing lived experiences shaped by socioeconomic status. Applying SCT in this context may elicit key factors for developing tailored interventions that better address the specific challenges experienced by this population. This study fills an important gap in the literature by exploring in-depth sociocultural and environmental factors that maintain smoking and make it difficult to quit in the context of experiencing a lower income.

## 2. Materials and Methods

### 2.1. Study Setting

Data for this study were collected from participants seeking care at safety-net primary care clinics within the San Francisco Health Network (SFHN) and other San Francisco Bay Area sites, which was part of a larger cross-sectional study [26]. Following approval from the University of California, San Francisco, and Palo Alto University IRBs, researchers completed targeted recruitment using a convenience sampling method [27].

### 2.2. Recruitment Procedures

Researchers used a combination of advertising (e.g., recruitment flyers posted in clinic waiting rooms) and attendance (e.g., clinic staff disseminating study flyers to patients in waiting rooms) recruitment strategies [28]. Most participants were recruited with assistance from the clinic staff, who informed participants about the study. The lead author (MTC) and members of the research team screened participants by phone to ensure that they met inclusion criteria: (1) low-income as defined by the poverty threshold for the San Francisco Bay Area [29], (2) at least 18 years of age, (3) currently smoke tobacco, and (4) thought about or intended to quit smoking within 30 days. To assess current tobacco use, the lead author and research team asked each participant if they currently used tobacco. If they said yes, participants were then asked if they were thinking or had the intention to quit smoking within 30 days. All participants in the present study met the criteria.

Eligible participants completed a short survey [26] and, on average, completed a 60 min (min = 48, max = 111) semi-structured interview. All interviews were conducted in English, and with two members of the research team present, either the lead author and a member of the research team or two members of the research team. Interviews were audio-recorded, while one of the interviewers recorded key points. Each study visit began with informed consent, demographics, a short survey, and a semi-structured interview on sociocultural contexts, attitudes, and behaviors regarding smoking and quitting. Participants were compensated with a $50 gift card for their time.

### 2.3. Interview Guide

The interview guide was developed using a multiphase approach. First, initial interview questions were drafted by the lead author and alongside a member of the research team with backgrounds in substance use (e.g., tobacco and alcohol use), mobile health technology use for underserved populations, and mental health. Additional interview questions were also drafted from the literature on low-income populations, tobacco disparities, chronic illnesses, and Social Cognitive Theory [7,21,25,30]. Second, the lead author presented the interview guide to a team of clinical research psychologists, which included the senior author (RFM), with expertise in tobacco use and cessation interventions, working with underserved populations (e.g., Spanish speaking, low-income), and cognitive behavioral therapies (CBT) for feedback. Third, the lead author and a member of the research team modified the interview questions based on the feedback received from the clinical research psychology team. Fourth, before finalizing the interview questions, the team of clinical research psychologists vetted the interview guide for final suggestions and improvements. The interview guide contained questions pertaining to sociocultural contexts, attitudes, and behaviors regarding smoking and quitting. Examples of interview questions and prompts are found in Table 1.

### 2.4. Analytic Strategy

A random sample of twenty interviews from the original sample [26] was transcribed verbatim by a team of graduate-level students. A random sample approach was implemented to minimize research team bias regarding participant data selection, ensuring diverse perspectives of participants [31]. Transcriptions were supervised by one of the co-authors (BP) with knowledge of qualitative work. Utilizing an inductive-deductive coding approach [32,33,34], interviews were analyzed by a multidisciplinary team from racial and ethnic diverse backgrounds. We implemented a thematic analytic approach guided by Braun and Clarke’s [35] six-step process: (1) familiarization of data, (2) generation of initial codes, (3) combining codes into themes, (4) reviewing themes, (5) defining and naming themes, and (6) reporting of findings. Four team members (coding team A; MTC, OFRP, SR, EH) independently read the transcripts to familiarize themselves with the data. Next, coding team A coded three interviews to establish the initial codebook, which included descriptions of codes. Coding team A met weekly over three months to discuss codes, coding, and discrepancies until a consensus on codes and the coding process was obtained. Once established, the codebook and process were discussed with the remaining team members to ensure understanding of codes, definitions, and coding procedures. Transcripts were randomly and equally distributed across five coders (coding team B; MTC, OFRP, SR, ARH, SM). Members of coding team B independently coded transcripts and met weekly to process coding and any discrepancies until consensus was obtained. After completing the coding process, the lead author and co-author (OFRP) reviewed coded interviews to ensure coding accuracy. Theme development and naming were also completed through a consensus process. Finally, as in the work of Rojas Perez and colleagues [36], a theme was considered when observed patterns emerged from more than half of the participants.

## 3. Results

### 3.1. Sample Characteristics

The analytic sample consisted of 20 self-identified low-income individuals between the ages of 24 and 70 years who smoked (M = 48.0, SD = 11.33). Most participants self-identified as male (75.0%, *n* = 15), with the rest identifying as female (25.0%, *n* = 5). Half of the sample self-identified as African American/Black (50.0%, *n* = 10), followed by White (30.0%, *n* = 6), Native Hawaiian/Pacific Islander (10.0%, *n* = 2), and Biracial (10.0%, *n* = 2). Additionally, a small number of participants self-identified as Latine/Hispanic (15.0%, *n* = 3). All participants noted some level of education, with eleventh grade being the average (SD = 3.13). More than half of the participants reported a yearly household income of less than $20,000 and being currently unemployed (70.0%, *n* = 14). Finally, most reported being single (65.0%, *n* = 13) and diagnosed with one or more mental and/or physical conditions (65.0%, *n* = 13). All participants reported at least one previous quit attempt (100%, *n* = 20). Participant demographics and smoking characteristics are described in Table 2.

### 3.2. Themes

Our analysis yielded six major themes and several subthemes that described sociocultural contexts, attitudes, and behaviors regarding smoking and quitting in a sample of low-income adults who smoke. The core tenets of SCT—personal factors, environment, and behaviors—served as the overarching foundation for the six themes (Figure 1), along with the context of being a low-income individual. The six themes were organized in the following order: (1) caught between health and tobacco use, (2) the nuances of context, (3) roadblocks to quitting, (4) motivation without movement, (5) a temporary escape, and (6) one size does not fit all.


**Theme 1: Caught Between Health and Tobacco Use**


The first major theme highlighted participants’ manifestation of tobacco use disorder. Despite being aware of tobacco-related health risks, participants continued to smoke, prioritizing immediate relief while consciously overlooking the negative impact on their health. For example, participants described the short-term and long-term health risks of smoking but were unable to stop themselves from having a cigarette. Martha (all names are pseudonyms), a 38-year-old cisgender woman who was unemployed, described how she continued to smoke despite being aware of the known health risks.


*“I don’t know how to explain that, but it’s just like, I have to have [cigarettes]. I know it’s bad for me. It can create cancer, health issues, and everything, but I’m just still stuck on smoking.”*


Additionally, participants reflected on their understanding, awareness, and knowledge of nicotine’s addictive nature. Through their unsuccessful quit attempts, participants reflected on the difficulties of overcoming nicotine dependence. Participants openly acknowledged navigating their nicotine dependence and demonstrated an awareness of it. For instance, Marco, a 58-year-old cisgender man, shared his experience with trying to quit and staying quit.


*“If somebody offers me one [cigarette]. Even if in my mind, I tell myself, okay, I’m not gonna smoke right now or I’m not gonna smoke anymore. And if somebody was to offer me one, I feel powerless to say no thanks…Or if I see one [cigarette] on the floor, it’s hard for me to just walk away. I mean, mentally, I’m aware [addictive nature]. And I think, no I don’t want to do it right? But I struggle with myself and the habit overcomes me to where I’m kinda like a slave [to cigarettes]. I gotta go get it.”*



**First Things First: The Morning Cigarette**


Participants also conveyed the urge to smoke a cigarette immediately upon waking. For participants, having a morning cigarette helped them prepare for the day and gave them relief and/or comfort. In addition, several participants described that smoking was closely intertwined with morning rituals, such as having a cigarette alongside coffee. For example, Mike, a 42-year-old cisgender male and a current college student, described his morning routine.


*“I wake up, smoke a couple of cigarettes, drink coffee. I prepare myself for the day. Get ready for work, then go to a morning meeting, and interact with the other individuals that are in my household. And then, once I go to work, I smoke before I get in, and before I get on transportation, and I smoke when I get off transportation.”*



**Theme 2: The Nuances of Context**


The association between smoking and socioenvironmental context was longstanding and multifaceted. Smoking behaviors were heavily influenced by modeling parental history of smoking and environments in which smoking was considered a social norm. Participants explained how they indirectly (e.g., observation) learned about cigarettes and smoking from their parent(s) and/or family members who smoked. Participants shared how early exposure to smoking influenced their smoking onset. For instance, Mike described who and what influenced his smoking behaviors.


*“My friends and I, we’d wait to take cigarettes from their moms, from their parents. And then you know, we’d see them doing it, so we thought why not us do it as well [smoke cigarettes].”*


Moreover, passive exposure to smoking within social communities (e.g., peer groups) throughout the lifespan (from adolescence into adulthood) influenced early smoking onset and the continuation of smoking behaviors to the present day. Participants also described how spending a considerable amount of time with peer groups where smoking was the norm, knowing friends who smoked, indirect peer pressure (e.g., being in spaces where people were smoking even without being offered a cigarette), and gaining social acceptance from people who smoked contributed to their smoking behaviors. For example, Lucila, a 51-year-old cisgender woman who was unemployed, shared how the importance of fitting in influenced her smoking onset.


*“My friends were doing it [smoking], and you know, I followed the crowd and I started doing it [smoking] at like 13 or 14.”*



**The Social Spark**


The subtheme of social smoking highlights another major dimension of the association between smoking and socioenvironmental context. Participants were highly aware of how socializing with people who smoked contributed to and reinforced smoking behaviors. Participants also used smoking to socialize and build communities, which increased social contact with others, prompting feelings of belonging. Smoking was considered a bridge for participants, as it connected them with people and communities. For instance, Nate, a 40-year-old cisgender gay man, described how smoking increased their connection with others.


*“Well, you know. Like I said, the majority of the smoking that I do is like out on social situations. Especially if we’re like out and having drinks…There’s kind of a social ritual to it. Like going outside to have a cigarette with a group. And especially if you’re kinda in a noisy place. It’s kinda nice, like you get to chat more. It’s a little bit more of an intimate thing.”*



**Corner Pocket**


Finally, participants experienced and were exposed to a significant number of tobacco retailers in their neighborhoods, which contributed to and reinforced their smoking behaviors. Socioenvironmental factors were discussed and described within the context of accessibility and greater exposure to tobacco retailers. Participants shared how they frequented their local retailer (often described as convenience stores) that sold inexpensive tobacco products. For example, Ruben, a 50-year-old unemployed cisgender man, described where in his neighborhood he would purchase his tobacco products.


*“Right in the [name of neighborhood store] because it’s cheaper [tobacco products]. I go to the [name of neighborhood store] and I buy a couple of tins or a couple of pouches. You can get a pouch for about 5 bucks, or I go to the tobacco store because you know they have express, or you can buy the brand [cigarettes] that you want. So, yea, I go to the corner tobacco stores or smoke shops.”*



**Theme 3: Roadblocks to Quitting**


Participants described the many barriers and difficulties they experience when trying to quit and stay quit. One of the major barriers to quitting was attributed to the influence of friends and family members who smoked during social activities. Participants expressed the difficulty of turning down the offering of a cigarette from a friend or family during periods of abstinence. In addition, most participants recalled how prior periods of abstinence were interrupted by peer pressure (either indirect or direct). For example, Ruben described how one of his many quit attempts was interrupted by his social environment.


*“But now I try and get a puff off someone else’s or share one. Because, like everywhere I am, everyone smokes. It’s like hard [to quit] because I try not to go out there too much [where people are smoking], or like I slowed down a lot actually. But still, I still smoke.”*


Additionally, the second major barrier participants described when trying to quit and stay quit was their nicotine dependence. Participants described their relentless urges and desires to smoke during periods of abstinence, which often were interrupted by socioenvironmental and/or psychological distress (resulting from stress, social inequities, life circumstances), cues triggering cravings. For instance, Jason, a 61-year-old cisgender man employed part-time, described why quitting was difficult for him.


*“I’ve been smoking for so long that it has become a part of my day. Because I go back to the first cigarette of the morning. And it’s like a ritual. Its like a habit. And if I don’t have that first cigarette in the morning, I just can’t seem to be able to function. And then not to mention the challenge to stopping [smoking cigarettes]. I don’t like the frustration and craving. You know, it just keeps me off balance.”*



**Theme 4: Motivation without Movement**


Participants expressed intrinsic motivation to quit and reported a history of quit attempts. Most participants’ greatest motivation to quit smoking was their current and future health. Participants expressed fear of what would happen to them if they continued smoking yet described quitting as a future goal rather than an immediate goal. For example, Tracy, a 61-year-old cisgender woman who was unemployed, shared her worries about future negative health impacts resulting from smoking.


*“Yeah, I think about that [quitting] because I think about my grandmother, you know. I was there when she had both of her open-heart surgeries, and I knew she smoked cigarettes and stuff, so I say, Lord, please don’t let me be that way. I don’t want no scar on my chest.”*


In addition, participants noticed being aware of how smoking negatively impacted their everyday health. For instance, Daniel, a 24-year-old cisgender male who was currently unemployed, reflected on how smoking was negatively impacting his physical activities.


*“Like I said, it gets in the way of working out a little bit. I can feel it [the effect of smoking] in my lungs when I run. I know eventually it’s gonna be a problem. So, I seek to quit one day.”*


To a lesser extent, other motivators to quit smoking included financial impact and the impact smoking had on the health of their communities (e.g., family, friends, peers).


**Theme 5: A Temporary Escape**


The fifth major theme detailed how smoking was utilized to cope with stressful life circumstances and/or socioeconomic disadvantages. Participants expressed how smoking provided them with a break from everyday responsibilities and challenges. Smoking helped participants feel at ease. For example, Amanda, a 38-year-old cisgender woman who was currently unemployed, described how smoking helped her manage everyday stressors.


*“I smoke when I hear problems with my kids or when bill collectors call me. Or when I’m overwhelmed, trying to get stuff done. It always helps.”*


Moreover, participants described how smoking helped them cope with anxiety, negative thoughts, and/or negative emotions. For instance, Sherry, a 57-year-old cisgender woman and current college student, described how smoking helped her cope with anxiety and depression.


*“[I smoke] when I am annoyed or when I’m feeling a little anxious and depressed.”*



**Theme 6: One Size Does Not Fit All**


The final major theme of the study detailed participants’ prior experience with evidence-based cessation interventions, which included nicotine-replacement therapies (NRTs), medication (e.g., bupropion), and smoking cessation groups. Participants described how the aforementioned cessation interventions temporarily helped them maintain periods of abstinence before relapsing. In addition, participants described how some evidence-based cessation interventions failed to meet their needs. For instance, Gabe, a 39-year-old cisgender man and currently unemployed, shared how the nicotine gum failed to treat his tobacco use.


*“The gums are horrible, and no, I haven’t tried the patch. You know, for me, it’s not really the physical addiction; it’s more of wanting to do something about the habit that I have built. For me, what is more profound is [the habit] than the actual physical addiction.”*


Additionally, most participants described past or current experience with cessation courses or groups but noted limited success and low engagement. For example, Jeffery, a 49-year-old cisgender man and currently unemployed, reflected on how cessation groups were unhelpful.


*“Counseling groups to me are like na, you know what I mean. Excuse the expression, but I think that they blow a lot of smoke up your butt.”*


## 4. Discussion

Drawing on emerging insights from in-depth interviews, the results from this study describe the complex and often subtle ways in which socioenvironmental contexts interact and influence attitudes and behaviors regarding smoking and quitting among low-income people. Participant narratives revealed the multifaceted nature of smoking, as individuals continued to smoke despite recognized health risks, with socioenvironmental factors playing a key role in both maintaining smoking and hindering cessation. Addiction unfolded in the context of ongoing life stressors, during social interaction, in environments where cigarettes were convenient and readily available, and through recurring urges and cravings, a habitual pattern that was difficult to change and recurrent relapse. These findings demonstrate how cigarette smoking persists among low-income marginalized communities, which may inform important implications for clinical care and highlight the importance of person-centered, tailored cessation approaches.

### 4.1. Interpretation

The lived experiences described in participant accounts underscore the social acceptability of smoking and the influence of pro-smoking neighborhood contexts. Early exposure to smoking through observing parents, relatives, and friends played a role in shaping initial smoking behaviors, often initiated during adolescence. These findings align with previous studies, which have found that parental smoking is a significant risk factor for smoking initiation among adolescent children [37]. Prior research has also concluded that one of the greatest influences on adult smoking stems from current close friends [38]. Notably, the majority of participants in this cohort did not report experiencing direct peer pressure to smoke; rather, their smoking initiation was often influenced by passive exposure to others who smoked. Aligned with SCT [39], which posits that behavior is shaped by various social contexts (such as family, peers, and community settings) participants were influenced by modeled behaviors within their environments, particularly from family members and peers.

Smoking was also described as a way to connect with others who smoke, promoting a sense of community and closeness. Despite conflicting evidence regarding the social nature of smoking, with some studies suggesting that smoking may contribute to increased loneliness and social isolation [40], it is crucial to account for the cultural and socioeconomic contexts in which smoking occurs. Evidence suggests that smoking tends to be a more frequent social activity among younger individuals belonging to lower-income strata [41]. Additionally, research has shown that familial interdependence is more prevalent in lower-income populations compared to middle- and higher-income populations, regardless of race and/or ethnicity [42,43]. These findings suggest that smoking may serve a social function among lower-income individuals, fostering social support and connectedness.

Participant accounts also underscored the convenience and accessibility of purchasing cigarettes within their local neighborhoods and communities. Many reported purchasing tobacco products from their neighborhood convenience stores and tobacco shops, often because the products were cheaper in these areas. Such accessibility and affordability of cigarettes in low-income neighborhoods may reinforce existing socioeconomic disparities and contribute to broader health disparities. These findings are consistent with previous research conducted throughout the United States, which has shown that cigarette prices tend to be lower in marginalized neighborhoods that have a greater percentage of youth, racial and ethnic minorities, and socioeconomically disadvantaged individuals [44,45]. Living in environments where smoking is both normalized and easily accessible can create confusion. On the one hand, individuals recognize the health risks of smoking. On the other hand, the widespread presence of smoking (especially in low-income neighborhoods) can reinforce the perception that smoking is routine. This dynamic can contribute to an internal conflict, where smoking is known to be harmful, yet simultaneously perceived as permissible because it is common and readily available.

Aligned with previous findings on the disconnect between the awareness of health risks and continued smoking [46], participants exhibited cognitive dissonance (i.e., smoking behavior contradicts knowledge of detrimental health outcomes). They acknowledged both the addictive nature of smoking and its associated health risks yet continued to smoke. Notably, and contrary to the smoking literature regarding cognitive dissonance, participants did not express unrealistic optimism about their chances to avoid smoking-related illnesses [47] and did not rationalize their smoking behavior [48]. Instead, participants acknowledged the difficulty in refusing cigarettes as they described their own personal and varying expressions of smoking. Participants also expressed smoking cessation as an important future consideration rather than an immediate decision, despite many reporting moderate motivation to quit and having made previous quit attempts. Motivation to quit was largely driven by observed—witnessing others endure tobacco-related health outcomes—or experienced health concerns, underscoring the continued influence of the surrounding socioenvironmental context.

Participants’ difficulty in prioritizing smoking cessation appeared to be influenced by their use of smoking as a coping strategy for stressful life circumstances. Previous studies suggest that individuals from low-income backgrounds exhibit greater smoking-related behavioral dependence in response to higher stress levels compared to those in higher income brackets [49]. Additionally, heightened perceived stress can maintain smoking behavior, particularly among low-income populations [49,50]. Prior research has also shown that low-income individuals manage life stressors (e.g., economic instability) with smoking [24]. Many individuals from low-income backgrounds often have limited resources for managing stress, likely due to social barriers [51,52]. Contributing factors include living in under-resourced neighborhoods, limited access to high-quality educational opportunities, and minimal exposure to mental health supports or wellness tools [30,52]. In these contexts, smoking serves as a conveniently available, socially acceptable, immediate—though short-term—relief from stress. In environments where smoking is often normalized and stress is chronic [53], smoking can become a persistent behavioral pattern to relieve stress, making cessation particularly difficult.

A noteworthy challenge discussed by participants that also hindered sustained cessation was the lack of success with treatments that did not resonate or work for them. While evidence-based smoking cessation approaches offered some temporary support during periods of abstinence, many participants reported frequent relapse and felt that these approaches did not adequately meet their needs. NRTs were often viewed as unhelpful, and while some individuals had tried medication or attended smoking cessation groups, these interventions were generally perceived as ineffective for long-term cessation. These findings reflect existing evidence that, while NRTs can be helpful, they are effective for only about 20% of individuals trying to quit [54]. Similarly, research indicates that behavioral interventions alone have success rates ranging from 7 to 16% [55]. Recent research conducted with a large, statewide sample in Missouri (U.S.) found that nearly 90% of low-income individuals who smoke had made at least one quit attempt. Over half of these individuals attempted to quit using nicotine patches, and 1 in 3 used nicotine gum [56]. Notably, rates for utilizing various cessation methods were found to be higher among low-income individuals than the general U.S. adult population [56]. Building on this previous work, our findings further demonstrate that this population is utilizing evidence-based cessation methods, making multiple attempts to quit, and continuing to relapse. These findings also underscore the substantial unmet need for cessation strategies that are tailored to the unique circumstances of low-income people who smoke. Addressing this health disparity requires the development and tailoring of smoking cessation methods that are not only evidence-based but informed by the lived experiences and realities of those most impacted.

The results from this study underscore that tobacco use is not a uniform experience, but rather one that manifests differently across individuals based on varying factors (e.g., socioenvironmental, personal, behavioral) and intersecting identities. Patient-centered approaches to care that encourage collaboration between patients and providers and emphasize personalized, tailored approaches to smoking cessation may improve quit rates among low-income people who smoke [57,58]. Existing research suggests that personalized approaches that identify patients’ social and behavioral needs and incorporate individualized action planning—extending beyond tobacco use—are viewed favorably by both patients and health educators (e.g., nurses) [59,60]. This is especially relevant among low-income people who smoke, given that socioenvironmental factors influence the onset and maintenance of smoking behavior.

### 4.2. Implications

Given the disproportionate health impacts of smoking among low-income populations—as well as the socioenvironmental factors that maintain smoking in these communities—a collaborative and multi-level response from healthcare providers, community stakeholders, and policymakers is essential.

#### 4.2.1. Practice Recommendations

Utilizing SCT, providers can address personal, behavioral, and environmental determinants in therapy that influence smoking to guide patients through the process of therapy. Integrating Motivational Interviewing (MI) + CBT provides a complementary approach that is person-centered and skills-based to enhance smoking cessation [61]. By implementing a person-centered (MI), educational and skills-based (CBT) therapeutic approach, providers can enhance an individual’s motivation and understanding of their smoking and address the underlying factors that contribute to current smoking beliefs and behaviors. For example, when working through cognitive dissonance and ambivalence (Motivation Without Movement and Caught Between Health and Tobacco Use), providers can implement microskills such as complex reflections (e.g., double-sided reflections) prior to engaging in and throughout cessation interventions, which may benefit those who feel both motivated and hesitant about quitting [62]. These reflections allow individuals to safely explore their ambivalence about smoking cessation, recognizing both reasons for continuing to smoke as well as reasons for quitting [62]. This approach not only normalizes the complexity of behavioral change but also provides the opportunity for healthcare educators and providers to have more in-depth conversations about overcoming barriers to quitting, while also enhancing the awareness of internal conflicts (e.g., valuing health yet continuing to smoke) [62,63].

Furthermore, when gathering information on maintaining factors of smoking and addressing barriers to quitting (The Nuances of Context and Roadblocks to Quitting), providers can integrate self-monitoring, which includes enhancing an individual’s self-awareness of smoking behaviors as well as their associated patterns, thoughts, emotions, situations, and environments [64]. By identifying patterns, providers and patients gain insight into smoking behavior, which then provides a starting point for how to identify triggers as they occur and how to manage triggers through developing coping strategies. Self-monitoring is an inherently tailored approach that gives patients the opportunity to lead therapeutic collaboration through providing clear points of intervention [65], which also provides patients a sense of control over their behavior [66].

Additionally, when addressing smoking as a coping mechanism for stressful life events (A Temporary Escape), providers can teach and practice stress management techniques with patients, such as progressive muscle relaxation, deep breathing, and behavioral activation. These techniques provide individuals with practical tools to manage stress in the moment by helping them manage the urge to smoke [67]. Pairing these skills with psychoeducation on nicotine dependence—particularly the understanding that cravings are temporary—and building self-awareness can further support patients in managing stress without turning to cigarettes.

Furthermore, evidence suggests that behavioral smoking cessation interventions work best when combined with pharmacotherapy (e.g., NRTs, Bupropion) [68]; however, results from this study indicated that the smoking cessation approaches participants utilized did not meet their needs for sustained cessation (One Size Does Not Fit All). Therefore, we strongly recommend that providers engage with patients utilizing a person-centered approach when providing psychoeducation on and discussing pharmacotherapy for smoking cessation. For example, providers may inquire about patient preferences for medication-based treatment, about which medications have or have not been effective in the past, and about their level of comfort with initiating pharmacotherapy. However, providers must allow adequate time for decision-making, recognizing that patients may require space to process the information. Also, implementing comprehensive psychoeducation that describes how to properly use evidence-based treatments like NRTs, may enhance both treatment adherence, health literacy, and successful cessation efforts.

#### 4.2.2. Community-Level and Broader Policy Recommendations

Provided that the findings from this study reveal the multifaceted influence of socioenvironmental factors on smoking and quitting (The Nuances of Context), recommendations for collective action are offered. Community health outreach [69] through distributing educational resources (e.g., pamphlets, flyers) across various settings—schools, churches, clinics, community recreational programs—in underserved neighborhoods can enhance community awareness of tobacco-related harms. Providing clear, jargon-free messaging about targeted marketing practices of tobacco companies, the influence of family and peers in smoking initiation and maintenance, and the health risks of smoking may encourage broader community dialog. Importantly, collaborative efforts between local and state officials, researchers, and healthcare professionals are essential for the development and dissemination of community educational materials.

For outreach programs to be feasible, accessible, and effective, funding mechanisms for community-based participatory research (CBPR) [70] must expand. Current smoking cessation treatments do not meet the needs of low-income people who smoke despite their quit attempt efforts [56]. Health-related intervention research that is informed by the communities of which they intend to serve is essential for developing effective and contextually relevant solutions. Findings from studies that incorporate CBPR methods have shown that CBPR better serves community objectives and can be an effective tool for policy change [71,72]. While federal funding mechanisms have recently undergone major financial cuts, state and local legislators have a critical opportunity to allocate resources toward addressing health disparities experienced by their constituents.

### 4.3. Strengths and Limitations

Applying SCT to qualitative design provides an in-depth theoretical approach to interpreting the study data, enhancing our understanding of the factors that contribute to smoking initiation, maintenance, cessation challenges, and relapse among low-income individuals who smoke. Findings from this study highlight the multifaceted role of socioenvironmental factors in shaping both smoking and quitting behavior, offering an important perspective of their impact on behavior change in the context of low-income. Future studies may explore the integration of peer support into smoking cessation interventions for low-income populations. Finally, given that participants frequently identified health-related motivations for quitting, incorporating physical activity into existing smoking cessation interventions may promote health and decrease smoking.

The study findings should be considered alongside study limitations. Participants were recruited from a large metropolitan city in the U.S. with considerable smoking bans and higher tobacco taxes [24]; therefore, the generalizability of these findings in more rural areas of the U.S. is unclear. To improve generalizability, future studies may benefit from including a more representative sample of low-income individuals who smoke across diverse geographic settings (e.g., urban vs. rural). Additionally, the interview protocol was developed via a Social Cognitive lens, potentially limiting our discovery of other salient factors that impact smoking and quitting among low-income populations. Future studies may consider incorporating other theories of behavior into interview protocol development to explore other factors not explored in the current study. The sample also consisted predominantly of men, limiting our ability to examine differing experiences between other gender identities. Further exploring intersecting identities may provide deeper insight into their role in sustaining smoking behavior.

## 5. Conclusions

This qualitative investigation revealed the complex and nuanced ways in which socioenvironmental contexts shape attitudes and behaviors related to smoking and quitting among low-income individuals. Grounded in SCT, participant accounts highlight how smoking is maintained through a broader interplay of personal (coping with stress, cognitive dissonance), behavioral (failed quit attempts, experience with evidence-based treatments), and environmental influences (neighborhood environment, family, peers), underscoring significant barriers to cessation. These findings highlight the potential value of peer-supported interventions that foster social connection and accountability in recovery. Additionally, MI + CBT may be a valuable approach in exploring quitting ambivalence and enhancing treatment engagement while learning skills for long-term cessation. Community health outreach via educational initiatives and calling on state and local governments to invest efforts in community-driven research is a step toward addressing health disparities that disproportionately impact low-income people who smoke. Taken together, the study findings point to the importance of patient-centered and collaborative approaches that tailor smoking cessation efforts to the unique needs and lived experiences of low-income people who smoke.

## Figures and Tables

**Figure 1 ijerph-22-01122-f001:**
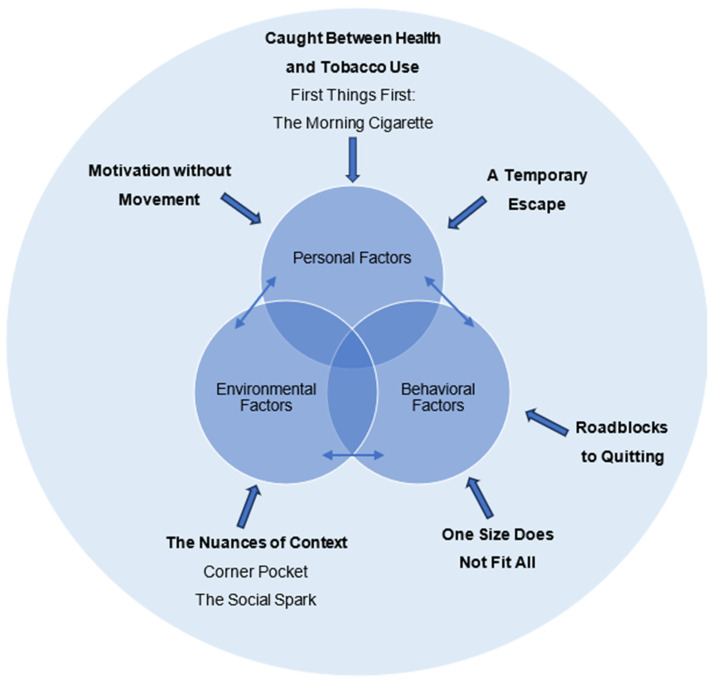
A visual representation of each qualitative theme mapped onto its corresponding SCT determinant.

**Table 1 ijerph-22-01122-t001:** Sample Interview Questions and Prompts.

When did you start smoking? Can you tell me more about that?
What was the main motivation for starting to smoke?
Where do you get your cigarettes from?
What do you like about smoking?
When and where do you usually smoke?
What kinds of situations make you want to have a cigarette? Can you give examples?
How has smoking affected your day-to-day life?
Do you have friends or family who smoke? How often do you see them? Do they offer you cigarettes? Have any of them tried to quit?
How does smoking make you feel?
Do you feel addicted to smoking? Can you tell me more about that?
What do you think about the health risks of smoking?
What are some benefits you see to smoking, if any?
Why do you want to quit smoking?
Have you tried to quit? Can you tell me more about that?
What have you tried to help you quit smoking?
What do you think is hard about quitting?
What are some challenges that may keep you from successfully quitting?

**Table 2 ijerph-22-01122-t002:** Participant demographics and smoking characteristics.

	Total *N* = 20
Demographics	
Gender *n* (%):	
Male	15 (75.0)
Female	5 (25.0)
Age (mean, SD)	48.0 (11.33)
Education (mean, SD)	11.15 (3.13)
Race *n* (%):	
African American/Black	10 (50.0)
Native Hawaiian/Pacific Islander	2 (10.0)
White	6 (30.0)
Biracial	2 (10.0)
Ethnicity *n* (%):	
Latinx/Hispanic	3 (15.0)
Not Latinx/Hispanic	10 (50.0)
Unknown/Not reported	7 (35.0)
Marital Status *n* (%):	
Single (never married)	13 (65.0)
Married, or in a domestic partnership	3 (15.0)
Separated	1 (5.0)
Divorced	2 (10.0)
Widowed	1 (5.0)
Employment Status *n* (%):	
Full-time	1 (5.0)
Part-time	2 (10.0)
Retired	1 (5.0)
Student	2 (10.0)
Unemployed	14 (70.0)
Smoking Characteristics	
Age of Smoking Onset *n* (%)	
Preteen	5 (25.0)
Teen	10 (50.0)
Post-teen adult	5 (25.0)
Previous Quit Attempt *n* (%)	
Yes	20 (100.0)
No	0 (0.0)
Reasons for Quitting *n* (%)	
Health	16 (80.0)
Finances	4 (20.0)
Community	2 (10.0)
Dislikes Smell	2 (10.0)
Did not want to quit	2 (10.0)

## Data Availability

The datasets presented in this article are not readily available because of confidentiality regarding the involvement of a vulnerable population. Requests to access the datasets should be directed to monique.cano@yale.edu.

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
