# Peer review of "Rethinking Smoking and Quitting in Low-Income Contexts: A Qualitative Analysis with Implications for Practice and Policy"

_ijerph, 2025, doi:10.3390/ijerph22071122_

Round 1
Reviewer 1 Report
Comments and Suggestions for Authors
Introduction:
The introduction is well-written and sets a clear rationale for the study.
The introduction could briefly outline what qualitative methods offer over quantitative surveys in this specific context.
Method:
Briefly write how selection bias was mitigated.
Clarify how many individuals were approached vs. how many participated to understand recruitment efficiency.
Line 144: It is mentioned that a sample of 20 interviews from the original sample of 25 was transcribed. Kindly clarify the difference of 5. Why was it not considered?
Shift lines 95 to 98 to the 2.2 section, as it speaks about the recruitment procedure.
Mention the average duration of the interview. 60 minutes may not be equal for all. You can include the minimum and maximum duration of interview as well.
Line 146: Did you use both the Inductive and Deductive approaches or any one? Need some clarity on the approach.
Analysis:
Clarify if qualitative software (e.g., NVivo, Atlas.ti) was used.
Although the paper is conceptually grounded in SCT, some thematic findings could be more explicitly linked back to SCT constructs. Consider a short table or diagram that maps the six themes to SCT components for clarity and theoretical alignment.
Discussion:
The discussion is lengthy and could be better structured using subheadings for “Interpretation,” “Implications,” and “Limitations.”
Author Response
Dear Reviewer,
We appreciate your helpful feedback and believe that the revisions we have made as a result of your suggestions have improved the quality of the manuscript. Below we have outlined the revision that we have made to the paper.
Introduction
Comment 1: The introduction is well-written and sets a clear rationale for the study.
Response 1: We appreciate the kind comment from the reviewer.
Comment 2: The introduction could briefly outline what qualitative methods offer over quantitative surveys in this specific context.
Response 2: We appreciate the suggestion. We added the following to the introduction (lines 80-83): “As a result, the research team deemed a qualitative approach to be suitable for the present study, given the methodology's nature to gain a deeper, more nuanced understanding of behavior and context [25], which is often limited in quantitative research.”
25. Patton, M. Q. Two Decades of Developments in Qualitative Inquiry: A Personal, Experiential Perspective. Qualitative Social Work 2002, 1 (3), 261–283. https://doi.org/10.1177/1473325002001003636.
Method
Comment 3: Briefly write how selection bias was mitigated.
Response 3: We appreciate the suggestion. We added the following to the Analytic Strategy subheading (lines 150-152): “A random sample approach was implemented to minimize research team bias regarding participant data selection, ensuring diverse perspectives of participants [31].”
Moser, A., & Korstjens, I. (2018). Series: Practical guidance to qualitative research. Part 3: Sampling, data collection and analysis. European journal of general practice, 24(1), 9-18.
Comment 4: Clarify how many individuals were approached vs. how many participated to understand recruitment efficiency.
Response 4: Thank you for this observation. The current random sample consists of participants from a larger outcome study (N = 64) who completed a self-report questionnaire and a semi-structured interview. The sample size for the outcome study was determined through a power analysis, which suggested a total sample size of between 42 and 102. Finally, given that we implemented a combination of advertising (e.g., recruitment flyers posted in clinic waiting rooms) and attendance (e.g., clinic staff disseminating study flyers to patients in waiting rooms), we are unable to accurately determine how many participants expressed interest and enrolled in the study due to the multiple recruitment strategies implemented.
Comment 5: Line 144: It is mentioned that a sample of 20 interviews from the original sample of 25 was transcribed. Kindly clarify the difference of 5. Why was it not considered?
Response 5: Thank you for this observation. A random sample of 20 was selected from the original dataset of N = 64 (see response above). The 25 being referenced correspond to the citation to the outcome study mentioned in the prior response. Due to edits, that citation number is now 31. Please see response 3 for justification of the random sample approach.
Comment 6: Shift lines 95 to 98 to the 2.2 section, as it speaks about the recruitment procedure.
Response 6: Thank you for the suggestion. We shifted lines 95 to 98 to section 2.2. recruitment procedures.
Comment 7: Mention the average duration of the interview. 60 minutes may not be equal for all. You can include the minimum and maximum duration of interview as well.
Response 7: Thank you for this observation. We have reworked the sentence and provided the minimum and maximum duration of interviews (lines 114-115): “Eligible participants completed a short survey and, on average, completed a 60-minute (min = 48, max = 111) semi-structured interview.”
Comment 8: Line 146: Did you use both the Inductive and Deductive approaches or any one? Need some clarity on the approach.
Response 8: Thank you for this observation. We selected an inductive-deductive approach because it allowed us to openly explore the data and connect findings to existing theory (social cognitive theory). Thus, leading to a more comprehensive and rigorous understanding of the data. The inductive-deductive approach offers flexibility, creating space for new insight and deeper understanding of the data while also taking relevant theory into account. We have included the following citation to lines 153-154:
Vanover, C., Mihas, P., & Saldaña, J. (Eds.). (2021). Analyzing and interpreting qualitative research: After the interview. Sage Publications.
Analysis
Comment 9: Clarify if qualitative software (e.g., NVivo, Atlas.ti) was used.
Response 9: This is a great question. Our team originally began coding on Dedoose, and quickly realized it was not for us. Therefore, we opted for the traditional method, which involved organizing and coding our data in Excel. One of our co-authors (our lead qualitative expert) has more than 12 years of experience organizing and coding data in Excel.
Comment 10: Although the paper is conceptually grounded in SCT, some thematic findings could be more explicitly linked back to SCT constructs. Consider a short table or diagram that maps the six themes to SCT components for clarity and theoretical alignment.
Response 10: Thank you for this excellent suggestion. We have now added a figure (Figure 1) that links our findings to SCT constructs.
Discussion
Comment 11: The discussion is lengthy and could be better structured using subheadings for “Interpretation,” “Implications,” and “Limitations.”
Response 11: Thank you for this wonderful suggestion. We have added subheadings to the discussion section.
Reviewer 2 Report
Comments and Suggestions for Authors
Dear Author(s),
My overall evaluation is that the paper has its merits, and its content might later constitute part of the official literature.
Nonetheless, since several major weaknesses remain, my recommendation is for a major revision.
While the study aims to explore the sociocultural contexts underlying smoking and cessation behaviors, it falls short in explicitly addressing any cultural characteristics of the low-income U.S. sample. The discussion lacks clarity regarding the specific cultural factors under consideration—whether these refer to cultural values, characteristics associated with low-income status, or the racial and ethnic backgrounds of the participants.
Although Social Cognitive Theory (SCT) offers a valuable framework for explaining how social learning and environmental interactions influence smoking and quitting behaviors, it has some limitations. Specifically, SCT does not adequately account for the complex dynamics and non-linear thinking processes involved in smoking behaviors—for instance, the contradiction between smokers’ awareness of health risks and their continued smoking. Furthermore, it offers a limited explanation for the internal updating mechanisms that enable successful cessation.
To address these gaps, I recommend the authors consider incorporating the Granular Interaction Thinking Theory (GITT). GITT complements SCT by emphasizing both individual and collective information processing, as well as the formation of values through social and environmental interactions. Quitting smoking is not solely a behavioral outcome but a value shift that depends on trusting reliable information and undergoing a transformative decision-making process. Such a process involves overcoming the acceptance–rejection threshold—a “point of no return”—at the individual level. A well-functioning information processing system can support smokers in internalizing the value of quitting and sustaining this behavioral change. For your reference, GITT:
- Vuong QH, Nguyen MH. (2024). Exploring the role of rejection in scholarly knowledge production: Insights from granular interaction thinking and information theory. Learned Publishing, 37(4), e1636. https://doi.org/10.1002/leap.1636
Among the six themes identified in the study, the second and sixth themes, in my view, highlight a critical aspect of both successful and unsuccessful smoking cessation efforts: the process is highly conditional on the interaction between the smoker’s internal information set (e.g., core values, knowledge, prior experiences) and external social-environmental information (e.g., observing peers, social interactions, availability and affordability of tobacco products). While the second theme is thoroughly discussed, the sixth theme receives limited elaboration and would benefit from deeper analysis. GITT can help elaborate on the sixth theme and its connection to the second theme.
Wishing you a productive and fruitful revision round. I look forward to reading your revised text in due course.
Best regards,
Author Response
Dear Reviewer,
We appreciate your helpful feedback and believe that the revisions we have made as a result of your suggestions have improved the quality of the manuscript. Below we have outlined the revision that we have made to the paper.
Comment 1: While the study aims to explore the sociocultural contexts underlying smoking and cessation behaviors, it falls short in explicitly addressing any cultural characteristics of the low-income U.S. sample. The discussion lacks clarity regarding the specific cultural factors under consideration—whether these refer to cultural values, characteristics associated with low-income status, or the racial and ethnic backgrounds of the participants.
Response 1: Thank you for this observation. We have added the following sentence to address the clarity regarding the cultural factor under consideration: “In the present study, the term 'sociocultural' is specific to a low-income context, encompassing lived experiences shaped by socioeconomic status.” Cultural factors such as specific cultural values, race and ethnicity, and specific cultural practices were not considered given that we recruited a fairly diverse sample (which we consider a strength of the study) and were focused on individuals who were living in poverty in an inner-city area.
Comment 2: Although Social Cognitive Theory (SCT) offers a valuable framework for explaining how social learning and environmental interactions influence smoking and quitting behaviors, it has some limitations. Specifically, SCT does not adequately account for the complex dynamics and non-linear thinking processes involved in smoking behaviors—for instance, the contradiction between smokers’ awareness of health risks and their continued smoking. Furthermore, it offers a limited explanation for the internal updating mechanisms that enable successful cessation.
To address these gaps, I recommend the authors consider incorporating the Granular Interaction Thinking Theory (GITT). GITT complements SCT by emphasizing both individual and collective information processing, as well as the formation of values through social and environmental interactions. Quitting smoking is not solely a behavioral outcome but a value shift that depends on trusting reliable information and undergoing a transformative decision-making process. Such a process involves overcoming the acceptance–rejection threshold—a “point of no return”—at the individual level. A well-functioning information processing system can support smokers in internalizing the value of quitting and sustaining this behavioral change. For your reference, GITT:
- Vuong QH, Nguyen MH. (2024). Exploring the role of rejection in scholarly knowledge production: Insights from granular interaction thinking and information theory. Learned Publishing, 37(4), e1636. https://doi.org/10.1002/leap.1636
Response 2: We appreciate the reviewer’s comments and share the view that smoking behavior involves complex psychological processes that are often non-linear. However, it is important to note that SCT does address the complex and non-linear cognitive dynamics involved in smoking behaviors. For example, Bandura (1978) states that, “Cognitive factors partly determine which external events will be observed, how they will be perceived, whether they have any lasting effects, what valence and efficacy they have, and how the information they convey will be organized for future use. The extraordinary capacity of humans to use symbols enables them to engage in reflective thought, to create and to plan foresightful courses of action in thought rather than having to perform possible options and suffer the consequences of thoughtless action. By altering their immediate environment, by creating cognitive self-inducements, and by arranging conditional incentives for themselves, people can exercise some influence over their own behavior. An act therefore includes among its determinants of self-produced influences.”
In the introduction we speak to the non-linear foundational components of SCT, which notes the reciprocal interactions between personal, behavioral, and environmental determinants. While we recognize the value of delving deeper into the cognitive processes (explaining cognitive dissonance via a SCT lens to highlight the complex and non-linear thinking processes that occur in conjunction with smoking behavior) encompassed by the personal determinant of SCT, such a detailed exploration of cognitive processes (e.g., decision making) falls beyond the aims and scope of the present study.
We appreciate the recommendation to incorporate the Granular Interaction Thinking Theory (GITT). However, our study was conceptualized and designed within a SCT framework. Introducing an additional theory would change the conceptualization, design, and purpose of the study and would extend the scope beyond our aims. Therefore, we maintain our focus on SCT as the foundation to the present study. In addition, we mention the following limitations in our manuscript (lines 602-606): “Additionally, the interview protocol was developed via a Social Cognitive lens, potentially limiting our discovery of other salient factors that impact smoking and quitting among low-income populations. Future studies may consider incorporating other theories of behavior into interview protocol development to explore other factors not explored in the current study.”
Comment 3: Among the six themes identified in the study, the second and sixth themes, in my view, highlight a critical aspect of both successful and unsuccessful smoking cessation efforts: the process is highly conditional on the interaction between the smoker’s internal information set (e.g., core values, knowledge, prior experiences) and external social-environmental information (e.g., observing peers, social interactions, availability and affordability of tobacco products). While the second theme is thoroughly discussed, the sixth theme receives limited elaboration and would benefit from deeper analysis. GITT can help elaborate on the sixth theme and its connection to the second theme.
Response 3: Thank you for the suggestion. We revisited the data and elaborated on theme six and provided additional quote that speaks to the data. The following was added (lines 381-387): “Additionally, most participants described past or current experience with cessation courses or groups but noted limited success and low engagement. For example, Jeffery, a 49-year-old cisgender man and currently unemployed, reflected on how cessation groups were unhelpful.
‘Counseling groups to me are like na, you know what I mean. Excuse the expression, but I think that they blow a lot of smoke up your butt.’”
Response to Quality of English Language: We acknowledge the reviewer’s suggestion regarding English language clarity. The manuscript has undergone multiple thorough revisions by all co-authors, who are native English speakers and are based in the United States; the manuscript has also been reviewed using professional editing tools to ensure grammatical accuracy and clarity. While we are confident that the language is appropriate for publication, we remain open to addressing specific suggestions for improvement should the reviewers and/or editors identify areas needing revision.
We sincerely appreciate the reviewer’s feedback and comments. We believe that with our new edits, the reviewers have helped us shape the manuscript into a stronger paper, and we are grateful for the reviewers’ contributions to this process. Thank you again for the opportunity to respond to reviewer feedback.